# Outcomes of Percutaneous Coronary Intervention in Elderly Patients with Rheumatoid Arthritis: A Nationwide Population-Based Cohort Study

**DOI:** 10.3390/healthcare11101381

**Published:** 2023-05-10

**Authors:** Bo Young Kim, HyeSung Moon, Sung-Soo Kim, Hyun-Sook Kim

**Affiliations:** 1Division of Rheumatology, Department of Internal Medicine, Gangneung Asan Hospital, University of Ulsan College of Medicine, Gangneung 25440, Republic of Korea; bo1156@ulsan.ac.kr (B.Y.K.); drkiss@ulsan.ac.kr (S.-S.K.); 2Rexsoft Corp., Seoul 08826, Republic of Korea; hsmoon9106@gmail.com; 3Division of Rheumatology, Department of Internal Medicine, Soonchunhyang University College of Medicine, Seoul 04401, Republic of Korea

**Keywords:** percutaneous coronary intervention, mortality, rheumatoid arthritis, elderly patients

## Abstract

Rheumatoid arthritis (RA) increases the risk of cardiovascular disease. This study aimed to evaluate the clinical outcomes of elderly patients with and without RA who underwent percutaneous coronary intervention (PCI). The Korean National Health Insurance Service claims database was used to extract data on 74,623 patients (14,074 with RA and 60,549 without RA) aged ≥ 65 years who were diagnosed with acute coronary syndrome and underwent PCI between 2008 and 2019. The primary outcome was survival of elderly patients with and without RA. The secondary outcome was survival in the RA subgroup. During a 10-year follow-up, the all-cause mortality survival rate was lower in patients with RA than that in patients without (53.7% vs. 58.3%, respectively, log-rank: *p* < 0.001). In the all-cause mortality RA subgroup, patients with elderly-onset RA had poor survival outcomes, whereas patients with young-onset RA had good survival outcomes compared with that in patients without RA (48.1% vs. 73.7% vs. 58.3%, respectively, log-rank: *p* < 0.001). Elderly patients with RA who underwent PCI had an increased mortality risk, particularly those with elderly rather than young-onset RA.

## 1. Introduction

Rheumatoid arthritis (RA) is a chronic inflammatory polyarthritis and is associated with an increased risk of cardiovascular diseases and mortality; patients with RA demonstrate a two-fold higher risk than the general population, similar to that in patients with type 2 diabetes [1,2]. 

With the increase in life expectancy, the proportion of elderly patients with RA is also increasing [3]. However, elderly patients frequently have comorbidities, and considering these adverse conditions, clinicians prefer less aggressive treatment in elderly patients with RA [4]. Recent studies demonstrated different features between elderly patients with elderly-onset RA and young-onset RA. Elderly-onset RA has a characteristic pattern with more comorbidities, a higher portion of male patients, acute onset, systemic involvement, and worse functional outcomes than young-onset RA [5,6]. These results suggest that there may be differences in the clinical characteristics and cardiovascular risk of elderly patients with RA, compared with those in the general elderly population, according to the age at RA onset. 

Percutaneous coronary intervention (PCI) is the most common revascularization method for patients with coronary artery disease. Studies have reported that even with acute coronary syndrome, typical angina symptoms are less frequent in patients with RA and general elderly patients [7,8]. These results suggest that elderly patients with RA undergoing PCI may experience delayed diagnoses and poor clinical outcomes compared with those in the general population. Despite the increasingly elderly population of patients with RA and the increased prevalence of patients with RA undergoing PCI, studies on PCI outcomes in elderly patients with RA are lacking [9,10,11,12]. Therefore, we aimed to evaluate the survival outcomes in elderly patients with RA in a large population-based PCI cohort over 12 years. We also analyzed the clinical factors affecting survival outcomes after PCI. 

## 2. Materials and Methods

### 2.1. Data Sources and Study Population

This retrospective cohort study was performed using data from the Korean National Health Insurance Service (KNHIS) database, which contains patient data including demographics, prescription records, procedure codes, and diagnosis codes of the International Classification of Diseases 10th Revision (ICD-10). 

We selected patients aged ≥ 65 years who were diagnosed with acute coronary syndrome and hospitalized with PCI procedure codes between 1 January 2008, and 31 December 2019 (Figure 1). Acute coronary syndrome is defined as the presence of ST-elevation myocardial infarction (STEMI), non-ST elevation myocardial infarction (NSTEMI), and unstable angina. We excluded patients with a diagnosis of autoimmune disease other than RA, those with a diagnosis of cancer, those hospitalized for MI or heart failure, and those who had undergone coronary artery bypass graft or PCI during the year before the PCI index date due to concerns that it may confound cardiac and all-cause mortality. The index PCI date was defined as the date with no coronary revascularization for at least one year before the first procedure date. Patients with RA were extracted based on ICD-10 diagnosis codes before the PCI index date. Patients diagnosed with RA before 2004 were excluded when considering the wash-out period for RA diagnosis because the KNHIS database was not properly established before 2003. Patients diagnosed with RA after the PCI index were also excluded. Consequently, 14,074 patients with RA were identified in the PCI cohort. 

### 2.2. Comorbidities 

Variables extracted as potential confounders included basic demographic characteristics, duration of RA, coronary artery disease at the PCI index date, number of PCI stents, and PCI stent type. Smoking status was categorized based on current smoking status (nonsmoker, ex-smoker, or current smoker). Alcohol abuse was defined as heavy drinking per the recommended standards of the National Institute on Alcohol Abuse and Alcoholism based on the patient’s standard drinking units [13]. Comorbidities were identified using diagnostic codes (ICD-10), such as hypertension, dyslipidemia, diabetes mellitus, atrial fibrillation, venous thromboembolism, peripheral vascular disease, stroke or transient ischemic attack (TIA), heart failure, chronic obstructive pulmonary disease (COPD), and moderate-to-severe chronic kidney disease (CKD). The prevalence of comorbidities was defined if the patient had a diagnosis code within 6 months of the PCI index date. Drug use was calculated for angiotensin-converting enzyme inhibitors or angiotensin receptor II blockers, beta-blockers, anticoagulants, antiplatelet agents, statins, and other lipid-lowering agents within 6 months of the PCI index date.

### 2.3. Study End-Points

We evaluated all-cause mortality and cardiovascular mortality of patients with and without RA who underwent PCI. We also assessed outcomes in patient subgroups with elderly-onset RA (presentation after the age of 65 years) and young-onset RA (presentation before the age of 65 years). In addition, clinical factors affecting all-cause and cardiovascular disease mortality were analyzed. All causes of mortality were defined using recorded information from 1 January 2008 to 31 December 2019 in connection with the Korean Statistical Information Service data where cardiac death is identified as the cause of death using the ICD-10 code for cardiovascular disease (I00-99).

### 2.4. Statistical Analysis

A *t*-test was used to compare continuous outcomes; the chi-squared test was used for categorical variable assessment between patients with and without RA. The Kaplan–Meier survival curves for patients with and without RA were plotted based on the covariates, and the log-rank test was used to confirm statistical significance. For multivariate analysis, multivariable logistic regression analysis was performed, adjusting for factors such as age, sex, current smoking, body mass index, stroke or TIA, heart failure, hypertension, dyslipidemia, diabetes mellitus, COPD, moderate-to-severe CKD, and diagnosis at PCI index date. Odds ratios (ORs) and 95% confidence intervals (CIs) were calculated using these models. 

*p*-values < 0.05 were considered statistically significant. The Statistical Package for SAS (version 9.4; SAS Institute Inc., Seoul, Republic of Korea) and R statistical software (version 4.0.3; R Foundation for Statistical Computing, Vienna, Austria) were used to perform the statistical analysis.

## 3. Results

### 3.1. Baseline Characteristics of the Study Participants

Table 1 summarizes the baseline characteristics of patients with and without RA. 

Compared with those without RA, patients with RA (n = 14,074) were older and included more women but were less likely to be current smokers. Patients with RA had frequent comorbidities; however, moderate-to-severe CKD rates were comparable between patients with and without RA. Among RA patients, the number of patients with elderly-onset RA (71.7%) was higher than that of patients with young-onset RA, and the mean duration of RA was 5.9 ± 3.5 years. 

Table 2 presents the procedural characteristics of the PCI index dates. Myocardial infarction (MI) comprised half the initial presentations, and STEMI was lower in patients with RA than in patients without RA. Most patients underwent PCI using a drug-eluting stent (DES) (96.9%) in a single vessel (87.8%). 

### 3.2. Association of RA and Clinical Outcomes in Elderly Patients Treated with PCI

Figure 2 shows the clinical outcomes of patients with and without RA who underwent PCI. All-cause mortality and cardiovascular disease-associated survival rates were lower in patients with RA than in patients without RA (log-rank: *p* < 0.001). During a follow-up period of 10 years, the all-cause mortality survival rates were 53.7% and 58.3% in patients with and without RA, respectively, whereas the cardiovascular disease mortality-associated survival rates were 79.4% and 81.1% in patients with and without RA, respectively.

Figure 3 shows the clinical outcomes of patients in the two RA subgroups. Compared to patients without RA, the survival rates associated with all-cause mortality and cardiovascular disease mortality were lower in patients with elderly-onset RA than in those with young-onset RA (log-rank: *p* < 0.001). During a 10-year follow-up period, the all-cause mortality-associated survival rates were 48.1%, 73.7%, and 58.3% in patients with elderly-onset RA, young-onset RA, and no RA, respectively. Furthermore, the cardiovascular disease mortality-associated survival rates were 75.9%, 90.5%, and 81.1% in patients with elderly-onset RA, young-onset RA, and no RA, respectively.

### 3.3. Clinical Factors Associated with the Survival Outcomes in Elderly Patients Treated with PCI

Table 3 and Table 4 show the results of multivariable logistic regression analyses regarding all-cause and cardiovascular mortalities. Compared to patients without RA, patients with RA, moderate-to-severe CKD (adjusted OR, 4.53; 95% CI, 3.90–5.28; *p* < 0.001), COPD (adjusted OR, 2.08; 95% CI, 1.96–2.20; *p* < 0.001), and diagnosis at PCI index date (adjusted OR of STEMI, 1.91; 95% CI, 1.84–1.98; *p* < 0.001) were at greater risk of all-cause mortality. In addition, age, male sex, current smoking, stroke/TIA, heart failure, hypertension, and diabetes mellitus were independent predictors of all-cause mortality (Table 3). Patients with RA diagnosed at PCI index date (adjusted OR of STEMI, 2.52; 95% CI, 2.40–2.66; *p* < 0.001; adjusted OR of NSTEMI, 1.51; 95% CI, 1.40–1.62; *p* < 0.001), stroke/TIA (adjusted OR, 1.75; 95% CI, 1.65–1.85; *p* < 0.001) and heart failure (adjusted OR, 1.55; 95% CI, 1.47–1.63; *p* < 0.001) were at greater risk of cardiovascular mortality; age, current smoking, hypertension, diabetes mellitus, COPD, and moderate-to-severe CKD were independent predictors of cardiovascular disease-associated mortality. Obesity was inversely associated with all-cause and cardiovascular disease-associated mortalities (adjusted OR, 0.94; 95% CI, 0.94–0.95; *p* < 0.001 for all-cause mortality; adjusted OR, 0.96; 95% CI, 0.95–0.97; *p* < 0.001 for cardiovascular mortality). Male sex was also inversely associated with cardiovascular mortality. 

In the RA subgroups, compared with those in the young-onset RA reference group, patients in the elderly-onset RA group had increased all-cause (adjusted OR, 1.80; 95% CI, 1.59–2.03; *p* < 0.001) and cardiovascular (adjusted OR, 1.36; 95% CI, 1.15–1.62; *p* < 0.001) mortalities (Table 4). Furthermore, patients with elderly-onset RA and significant associations with moderate-to-severe CKD (adjusted OR, 4.84; 95% CI, 3.44–6.85; *p* < 0.001), diagnosis at PCI index date (adjusted OR of STEMI, 2.28; 95% CI, 2.09–2.49; *p* < 0.001), and COPD (adjusted OR, 1.85; 95% CI, 1.63–2.11; *p* < 0.001) had increased all-cause mortality. Age, male sex, current smoking, stroke/TIA, heart failure, and diabetes mellitus were independent predictors of all-cause mortality. In addition, patients with elderly-onset RA and significant associations with diagnosis at PCI index date (adjusted OR of STEMI, 3.05; 95% CI, 2.71–3.43; *p* < 0.001; adjusted OR of NSTEMI, 1.77; 95% CI, 1.51–2.08; *p* < 0.001), stroke/TIA (adjusted OR, 1.69; 95% CI, 1.49–1.92; *p* < 0.001), and heart failure (adjusted OR, 1.49; 95% CI, 1.33–1.67; *p* < 0.001) had increased cardiovascular mortality. Age, current smoking, dyslipidemia, diabetes mellitus, and COPD were independent predictors of cardiovascular mortality. Obesity was inversely associated with all-cause and cardiovascular mortalities (adjusted OR, 0.94; 95% CI, 0.92–0.95; *p* < 0.001 for all-cause mortality; adjusted OR, 0.95; 95% CI, 0.94–0.97; *p* < 0.001 for cardiovascular mortality). Male sex was also inversely associated with cardiovascular mortality in patients with elderly-onset RA.

## 4. Discussion

This study shows that RA affects the survival outcomes of elderly patients with ischemic heart disease who undergo PCI. The all-cause and cardiovascular disease-associated survival rates were lower in patients with RA than in those without RA. In the RA subgroup analysis, the all-cause and cardiovascular disease-associated survival rates were lower in patients with elderly-onset RA but higher in those with young-onset RA, compared with those in the non-RA group. Additionally, our study identifies differences in patient characteristics and clinical factors associated with mortality between patients with and without RA who underwent PCI. 

The increased cardiovascular risk in RA due to premature atherosclerosis is associated with traditional cardiovascular risk factors and factors associated with RA, such as chronic inflammation and medication (including glucocorticoids and nonsteroidal anti-inflammatory drugs) [14,15,16,17,18]. In particular, chronic inflammation due to RA alters body composition, insulin sensitivity, and lipid profiles, and these metabolic outcomes lead to a metabolic syndrome that mediates premature atherosclerosis [19]. Moreover, age affects treatment patterns, resulting in less aggressive treatment in elderly patients with RA [4,20]. This treatment tendency in elderly patients with RA is related to the high disease activity of RA, which may lead to a poor prognosis in acute coronary syndrome [21,22]. In addition, the presence of comorbidities is associated with higher disease activity in RA and reduced medication adherence in patients with RA [23,24]. In our study, patients with RA had frequent comorbidities, reflecting an association of increased mortality with acute coronary syndrome. As regards time of RA diagnosis, elderly-onset RA tends to have a higher proportion of male patients, more acute onset, and greater systemic involvement than young-onset RA [5]. According to previous studies, patients with late-onset RA also have frequent comorbidities and higher disease activity of RA, which are associated with a poor prognosis in acute coronary syndrome [6,25]. Consistent with these findings, the all-cause and cardiovascular disease-associated survival rates in our study were lower in the late-onset RA group compared with those in the non-RA and early-onset RA groups. Late-onset RA was also an independent predictor of increased mortality compared with early-onset RA. Therefore, the larger late-onset RA subgroup may result in the lower survival rates of RA patients than the non-RA group.

Although most studies have shown consistent results of increased cardiovascular risk in patients with RA, numerous studies on PCI outcomes in patients with RA have shown controversial results. Furthermore, studies in populations of elderly patients with RA are lacking. A Japanese cohort study conducted in the general population (562,640 patients) of patients aged ≥ 60 years who had undergone PCI reported that older patients had a greater mortality risk compared with younger patients [26]. Another cohort study showed that the association between age and mortality rate after PCI ranges from 0.5% for patients < 55 years to nearly 5% for patients > 85 years [27]. There are several causes for poor outcomes after PCI in elderly patients. The elderly often present with atypical symptoms, resulting in delayed or missed diagnoses. Further, elderly patients demonstrate higher operative risk due to significant coronary artery disease burden and existing comorbidities [8,28]. Elderly patients with RA who have age-related increases in levels of proinflammatory cytokines and C-reactive protein, and higher disease activity of RA contributing to the inflammatory condition, may have a greater risk of mortality than the general elderly population [29]. In previous studies on PCI outcomes in patients with RA, RA was significantly associated with an increased risk of long-term mortality, consistent with the results of our study [10,30]. These studies not only involved elderly patients with RA but also included 80% of participants over 60 years of age. Risk factors for each PCI outcome, such as old age and RA, may synergistically affect elderly patients with RA, worsening survival outcomes. In our study, it is imperative to note that the PCI outcome in patients with RA was poor, despite the low rates of MI, compared with that in patients without RA at the PCI index date. The major clinical factors associated with mortality in our study were COPD, CKD, and STEMI diagnosis at PCI index date, in addition to the traditional risk factors involved in the Framingham Risk Score. Factors such as CKD and STEMI in particular have been previously identified as predictors of mortality after PCI in elderly patients [26,31]. Conversely, obesity was a factor that reduced mortality in our study because a low or normal weight, indicative of a state of rheumatoid cachexia, may be associated with an increased risk of mortality due to muscle loss caused by inflammatory cytokines in RA [30].

The strengths of our study include the evaluation of a large sample population (14,074 elderly patients with RA), achieved by using a large population-based medical records database. This overcame the low prevalence of RA and the difficulty of enrolling elderly patients in clinical trials. Further, we showed the survival outcomes in elderly RA patients after classification into elderly-onset and young-onset subgroups. To our knowledge, this is the first study to demonstrate clinical outcomes of elderly patients with RA undergoing PCI for acute coronary syndrome, reflecting the trends of an increasingly elderly population of patients with RA and the increasing prevalence of patients with RA undergoing PCI [9,10,11,12]. Second, we excluded patients with potential confounders, such as those hospitalized for cancer, MI, heart failure, or coronary revascularization during the one-year pre-PCI index period. Third, our study also demonstrated that 96.9% of patients underwent PCI with a DES. Beyond the era of balloon angioplasty and bare metal stents, DES use has become a standard strategy for PCI [32,33]. A high percentage of patients in our study underwent PCI with a DES, reflecting the current trend in standard treatment procedures that can appropriately evaluate PCI outcomes. Our study has a few limitations. First, since this was an observational study, the results should be cautiously interpreted in real-world scenarios. Further prospective studies may be needed to confirm the causal relationship between elderly patients with RA and PCI outcomes. Second, although we adjusted covariate imbalances by conducting multivariable logistic regression analysis, the possibility of unmeasured potential confounding variables cannot be ruled out. Third, this study lacked data on RA disease activity, RA drugs, and coronary lesions or complexity, which are known to affect cardiovascular outcomes. Fourth, data were not collected on activities of daily living, cognitive impairment, financial status, and overall frailty that may be required for cardiovascular care in elderly patients [34].

## 5. Conclusions

Elderly patients aged ≥ 65 years with RA who underwent PCI had an increased risk of all-cause and cardiovascular mortalities; this trend was more specifically observed in patients with elderly-onset RA who presented after the age of 65 years than in those with young-onset RA who presented before the age of 65 years. Patient characteristics and clinical factors associated with survival outcomes need to be considered during the treatment of elderly patients with RA for the better prognosis of cardiovascular diseases. 

## Figures and Tables

**Figure 1 healthcare-11-01381-f001:**
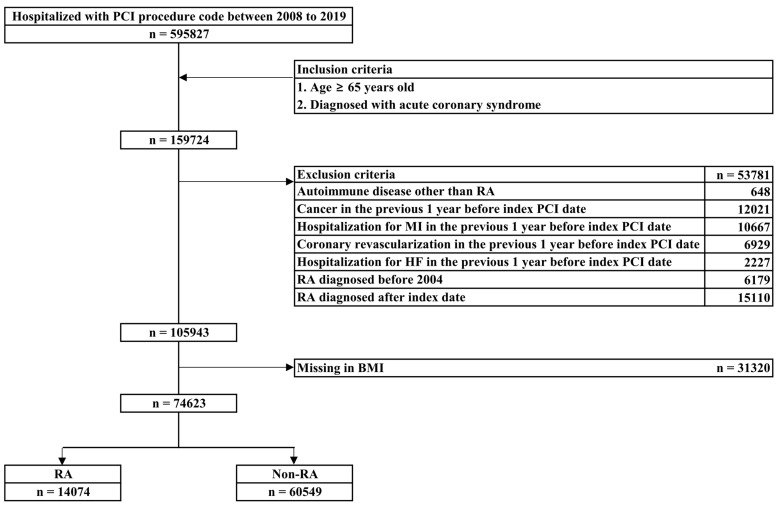
Flow chart of the elderly study population who underwent PCI between 2008 and 2019 in the Korean National Health Insurance Service database. PCI, percutaneous coronary intervention; RA, rheumatoid arthritis; MI, myocardial infarction; HF, heart failure; BMI, body mass index.

**Figure 2 healthcare-11-01381-f002:**
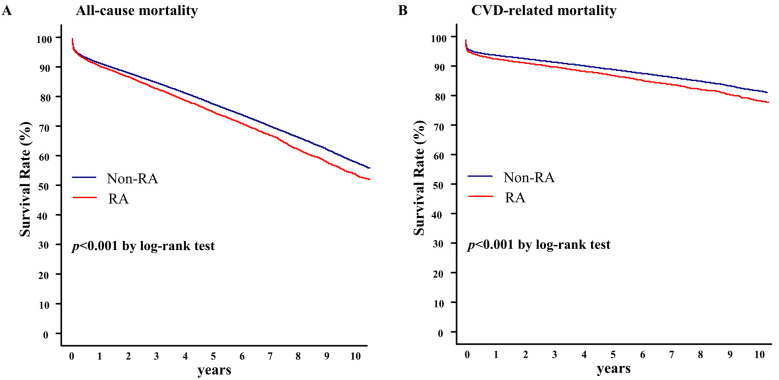
All-cause (**A**) and cardiovascular (**B**) mortality-associated survival rates in patients with and without RA. CVD, cardiovascular disease; RA, rheumatoid arthritis.

**Figure 3 healthcare-11-01381-f003:**
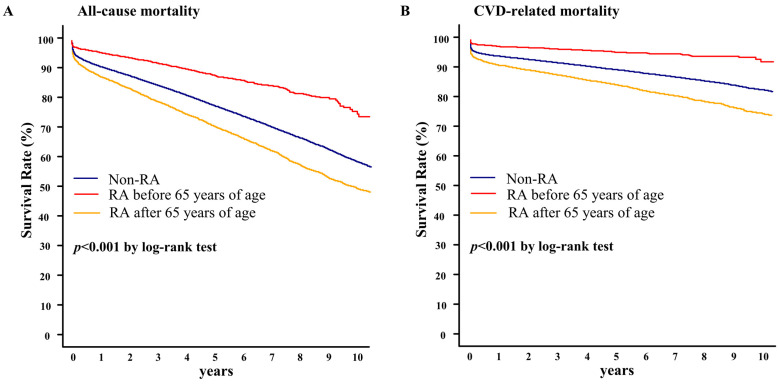
All-cause (**A**) and cardiovascular (**B**) mortality-associated survival rates in RA subgroups. CVD, cardiovascular disease; RA, rheumatoid arthritis.

**Table 1 healthcare-11-01381-t001:** Baseline characteristics of the included patients according to RA status.

Variables	Total	RA	Without RA	*p*-Value
	(n = 74,623)	(n = 14,074)	(n = 60,549)	
Age at index date (years), mean ± SD	73.7 ± 6.0	74.6 ± 6.1	73.5 ± 6.0	<0.001
Age at index date, n (%)				<0.001
65 to 69 years	21,644 (29.0)	3389 (24.1)	18,255 (30.2)	
70 to 79 years	39,764 (53.3)	7630 (54.2)	32,134 (53.1)	
≥80 years	13,215 (17.7)	3055 (21.7)	10,160 (16.8)	
Sex (women), n (%)	30,084 (40.3)	7791 (55.4)	22,293 (36.8)	<0.001
BMI (kg/㎡), mean ± SD	24.2 ± 3.1	24.4 ± 3.2	24.1 ± 3.0	<0.001
Current smoking, n (%)	14,881 (19.9)	2015 (14.3)	12,866 (21.3)	<0.001
Alcohol abuse, n (%)	249 (0.3)	59 (0.4)	190 (0.3)	0.061
Age at the time of diagnosis of RA, n (%)				<0.001
<65 years	-	3987 (28.3)	-	
≥65 years	-	10,087 (71.7)	-	
Duration of RA (year), n (%)	-	5.9 ± 3.5	-	<0.001
<1	-	1079 (7.7)	-	
1 to 5	-	5023 (35.7)	-	
>5	-	7972 (56.6)	-	
Comorbidities, n (%)				
Hypertension	64,269 (86.1)	12,449 (88.5)	51,820 (85.6)	<0.001
Dyslipidemia	16,565 (22.2)	3350 (23.8)	13,215 (21.8)	<0.001
Diabetes mellitus	40,307 (54.0)	8208 (58.3)	32,099 (53.0)	<0.001
Atrial fibrillation	5820 (7.8)	1156 (8.2)	4664 (7.7)	0.044
Venous thromboembolism	1841 (2.5)	425 (3.0)	1416 (2.3)	<0.001
Peripheral vascular disease	11,696 (15.7)	2744 (19.5)	8952 (14.8)	<0.001
stroke/TIA	10,887 (14.6)	2247 (16.0)	8640 (14.3)	<0.001
heart failure	14,851 (19.9)	3108 (22.1)	11,743 (19.4)	<0.001
COPD	5943 (8.0)	1221 (8.7)	4722 (7.8)	0.001
Moderate-to-severe CKD	823 (1.1)	165 (1.2)	658 (1.1)	0.406
Use of drugs, n (%)				
ACEI/ARB	11,883 (15.9)	2318 (16.5)	9565 (15.8)	0.051
Beta-blockers	57,072 (76.5)	10,659 (75.7)	46,413 (76.7)	0.021
Anticoagulants	3509 (4.7)	744 (5.3)	2765 (4.6)	<0.001
Antiplatelets	74,545 (99.9)	14,057 (99.9)	60,488 (99.9)	0.604
Statins	66,971 (89.8)	12,879 (91.5)	54,092 (89.3)	<0.001
Other lipid-lowering agents	4226 (5.7)	865 (6.2)	3361 (5.6)	0.006

Data are reported as mean ± SD for continuous variables. *p*-values were computed using Student’s *t*-test or Wilcoxon’s rank-sum test for continuous variables and the chi-squared test or Fisher’s exact test for categorical variables. RA, rheumatoid arthritis; SD, standard deviation; BMI, body mass index; TIA, transient ischemic attack; COPD, chronic obstructive pulmonary disease; CKD, chronic kidney disease; ACEI, angiotensin-converting enzyme; ARB, angiotensin receptor blocker.

**Table 2 healthcare-11-01381-t002:** Procedural characteristics of the included patients according to RA status at PCI index date.

Variable	Total	With RA	Without RA	*p*-Value
	(n = 74,623)	(n = 14,074)	(n = 60,549)	
Diagnosis at PCI index date, n (%)				<0.001
STEMI	28,662 (38.4)	4925 (35.0)	23,737 (39.2)	<0.001
NSTEMI	10,009 (13.4)	1996 (14.2)	8013 (13.2)	0.003
Unstable angina	35,952 (48.2)	7153 (50.8)	28,799 (47.6)	<0.001
Type of implanted stents *, n (%)				0.010
Drug-eluting stents	72,341 (96.9)	13,659 (97.1)	58,682 (96.9)	0.418
Bare metal stents	1282 (1.7)	216 (1.5)	1066 (1.8)	0.069
Both drug-eluting and bare metal stents	265 (0.4)	38 (0.3)	227 (0.4)	0.071
Number of implanted stents *, n (%)				0.056
single-vessel PCI	65,536 (87.8)	12,378 (88.0)	53,158 (87.8)	0.620
multi-vessel PCI	8352 (11.2)	1535 (10.9)	6817 (11.3)	0.239

*p*-values were computed using the chi-squared test or Fisher’s exact test. PCI, percutaneous coronary intervention; RA, rheumatoid arthritis; STEMI, ST-elevation myocardial infarction; NSTEMI, non-ST-elevation myocardial infarction. * The type and number of implanted stents were unknown in 735 (1.0%) and 735 (1.0%) patients, respectively.

**Table 3 healthcare-11-01381-t003:** Clinical factors associated with mortality in patients with and without RA.

Variable	Univariable Models		Multivariable Models	
	Crude OR (95% CI)	*p*-Value	Adjusted OR (95% CI)	*p*-Value
**All-cause mortality**				
Age (at index date)	1.11 (1.108–1.115)	<0.001	1.11 (1.106–1.113)	<0.001
Male	1.06 (1.02–1.09)	0.001	1.17 (1.12–1.21)	<0.001
Current smoking	1.30 (1.25–1.35)	<0.001	1.46 (1.40–1.53)	<0.001
Obesity	0.91 (0.91–0.92)	<0.001	0.94 (0.94–0.95)	<0.001
Stroke/TIA	1.85 (1.77–1.93)	<0.001	1.72 (1.64–1.80)	<0.001
Heart failure	1.65 (1.59–1.72)	<0.001	1.28 (1.23–1.34)	<0.001
Hypertension	1.21 (1.16–1.27)	<0.001	1.13 (1.07–1.19)	<0.001
Dyslipidemia	0.93 (0.89–0.97)	<0.001	0.99 (0.95–1.03)	0.637
Diabetes mellitus	1.42 (1.37–1.47)	<0.001	1.46 (1.41–1.51)	<0.001
COPD	2.60 (2.46–2.74)	<0.001	2.08 (1.96–2.20)	<0.001
Moderate-to-severe CKD	4.20 (3.65–4.84)	<0.001	4.53 (3.90–5.28)	<0.001
Diagnosis at PCI index date				
Unstable angina (reference)	1.00		1.00	
STEMI	2.14 (2.07–2.22)	<0.001	1.91 (1.84–1.98)	<0.001
NSTEMI	1.44 (1.37–1.51)	<0.001	1.13 (1.07–1.19)	<0.001
**CVD-related mortality**				
Age (at index date)	1.10 (1.095–1.102)	<0.001	1.09 (1.08–1.09)	<0.001
Male	0.85 (0.82–0.89)	<0.001	0.93 (0.88–0.97)	0.003
Current smoking	1.14 (1.08–1.20)	<0.001	1.29 (1.22–1.37)	<0.001
Obesity	0.93 (0.92–0.93)	<0.001	0.96 (0.95–0.97)	<0.001
Stroke/TIA	1.86 (1.76–1.96)	<0.001	1.75 (1.65–1.85)	<0.001
Heart failure	1.95 (1.85–2.04)	<0.001	1.55 (1.47–1.63)	<0.001
Hypertension	1.16 (1.09–1.24)	<0.001	1.09 (1.02–1.17)	0.011
Dyslipidemia	1.00 (0.95–1.06)	0.870	1.03 (0.97–1.09)	0.297
Diabetes mellitus	1.30 (1.25–1.36)	<0.001	1.28 (1.22–1.34)	<0.001
COPD	1.89 (1.77–2.02)	<0.001	1.49 (1.39–1.60)	<0.001
Moderate-to-severe CKD	1.61 (1.34–1.91)	<0.001	1.49 (1.23–1.79)	<0.001
Diagnosis at PCI index date				
Unstable angina (reference)	1.00		1.00	
STEMI	2.84 (2.71–2.99)	<0.001	2.52 (2.40–2.66)	<0.001
NSTEMI	1.86 (1.74–2.00)	<0.001	1.51 (1.40–1.62)	<0.001

Odds ratios were determined using multivariable logistic regression analysis for non-RA versus RA groups. The adjusted model includes age, sex, current smoking, body mass index, stroke/TIA, heart failure, hypertension, dyslipidemia, diabetes mellitus, COPD, moderate-to-severe CKD, and diagnosis at PCI index date. RA, rheumatoid arthritis; OR, odds ratio; CI, confidence interval; TIA, transient ischemic attack; COPD, chronic obstructive pulmonary disease; CKD, chronic kidney disease; PCI, percutaneous coronary intervention; STEMI, ST-elevation myocardial infarction; NSTEMI, non-ST elevation myocardial infarction; CVD, cardiovascular disease. The reference group for each odds ratio is patients without RA.

**Table 4 healthcare-11-01381-t004:** Clinical factors associated with mortality in subgroups of patients with RA.

Variable	Univariable Models		Multivariable Models	
	Crude OR (95% CI)	*p*-Value	Adjusted OR (95% CI)	*p*-Value
**All-cause mortality**				
Age at diagnosis of RA				
···Young-onset RA (reference)	1.00		1.00	
···Elderly-onset RA	3.35 (3.03–3.70)	<0.001	1.80 (1.59–2.03)	<0.001
Age (at index date)	1.10 (1.10–1.11)	<0.001	1.07 (1.06–1.08)	<0.001
Male	1.20 (1.11–1.29)	<0.001	1.19 (1.09–1.30)	<0.001
Current smoking	1.30 (1.25–1.35)	<0.001	1.46 (1.40–1.53)	<0.001
Obesity	0.91 (0.89–0.92)	<0.001	0.94 (0.92–0.95)	<0.001
Stroke/TIA	1.77 (1.61–1.95)	<0.001	1.65 (1.49–1.83)	<0.001
Heart failure	1.51 (1.39–1.64)	<0.001	1.19 (1.08–1.31)	<0.001
Hypertension	1.19 (1.06–1.34)	0.004	1.10 (0.97–1.26)	0.145
Dyslipidemia	1.01 (0.92–1.10)	0.873	1.05 (0.96–1.16)	0.288
Diabetes mellitus	1.31 (1.21–1.41)	<0.001	1.37 (1.26–1.49)	<0.001
COPD	2.37 (2.11–2.67)	<0.001	1.85 (1.63–2.11)	<0.001
Moderate-to-severe CKD	4.08 (2.99–5.62)	<0.001	4.84 (3.44–6.85)	<0.001
Diagnosis at PCI index date				
Unstable angina (reference)	1.00		1.00	
STEMI	2.62 (2.41–2.84)	<0.001	2.28 (2.09–2.49)	<0.001
NSTEMI	1.63 (1.45–1.82)	<0.001	1.31 (1.16–1.48)	<0.001
**CVD-related mortality**				
Age at diagnosis of RA				
···Young-onset RA (reference)	1.00		1.00	
···Elderly-onset RA	2.67 (2.33–3.08)	<0.001	1.36 (1.15–1.62)	<0.001
Age (at index date)	1.09 (1.08–1.10)	<0.001	1.06 (1.05–1.07)	<0.001
Male	0.90 (0.82–1.00)	0.046	0.89 (0.79–0.99)	0.040
Current smoking	1.18 (1.03–1.35)	0.016	1.33 (1.14–1.55)	<0.001
Obesity	0.92 (0.91–0.94)	<0.001	0.95 (0.94–0.97)	<0.001
Stroke/TIA	1.77 (1.57–2.00)	<0.001	1.69 (1.49–1.92)	<0.001
Heart failure	1.86 (1.66–2.06)	<0.001	1.49 (1.33–1.67)	<0.001
Hypertension	1.07 (0.91–1.26)	0.409	0.99 (0.83–1.17)	0.860
Dyslipidemia	1.15 (1.03–1.29)	0.013	1.15 (1.02–1.29)	0.025
Diabetes mellitus	1.23 (1.11–1.36)	<0.001	1.25 (1.12–1.39)	<0.001
COPD	1.85 (1.59–2.15)	<0.001	1.47 (1.25–1.72)	<0.001
Moderate-to-severe CKD	1.36 (0.88–2.02)	0.149	1.36 (0.86–2.08)	0.173
Diagnosis at PCI index date				
Unstable angina (reference)	1.00		1.00	
STEMI	3.51 (3.13–3.94)	<0.001	3.05 (2.71–3.43)	<0.001
NSTEMI	2.17 (1.82–2.53)	<0.001	1.77 (1.51–2.08)	<0.001

Odds ratios were determined using multivariable logistic regression analysis for elderly-onset RA versus young-onset RA groups. The adjusted model includes age, sex, current smoking, body mass index, stroke/TIA, heart failure, hypertension, dyslipidemia, diabetes mellitus, COPD, moderate-to-severe CKD, and diagnosis at PCI index date. RA, rheumatoid arthritis; OR, odds ratio; CI, confidence interval; TIA, transient ischemic attack; COPD, chronic obstructive pulmonary disease; CKD, chronic kidney disease; PCI, percutaneous coronary intervention; STEMI, ST-elevation myocardial infarction; NSTEMI, non-ST elevation myocardial infarction; CVD, cardiovascular disease. The reference group for each odds ratio is patients with RA onset before 65 years.

## Data Availability

The data generated and analyzed in this study are available from the Korean National Health Insurance Service at https://nhiss.nhis.or.kr. (accessed on 1 June 2022).

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
