# Peer review of "Outcomes of Percutaneous Coronary Intervention in Elderly Patients with Rheumatoid Arthritis: A Nationwide Population-Based Cohort Study"

_healthcare, 2023, doi:10.3390/healthcare11101381_

Round 1

Reviewer 1 Report

The authors demonstrated clinical outcomes of elderly patients with RA undergoing PCI for acute coronary syndrome in a large population sample and compared elderly- and young- onset RA subgroups.

Comments

1.      Lines 63-64; 117-120; All the repeated sentences should be removed.

2.      Table 1 and 2: p values should be placed in appropriate lines/

3.      Lines 130-132: The authors should describe which patients were considered  with elderly onset and which, with early onset here. Description of clinical characteristics of patients with RA are badly required. This should be corrected.

4.      Lines 240-245: The authors should not repeat their Results at Discussion section. This should be corrected.

5.      Lines 269-272: This sentence contains contradictory statements. This should be corrected.

6.      Lines 244-247: This data is related to Introduction section. This should be corrected.

Author Response

Thank you for your comments.

We did our best to respond to your comments.

All changes are marked in red in the revised manuscript.

Reviewer 1's comments  

  1. Lines 63-64; 117-120; All the repeated sentences should be removed.

Answer>

The sentences you pointed out has been deleted (part of 2.1 Data source and study population and part of 3.1 Baseline characteristics of the study participants)

  1. Table 1 and 2: p values should be placed in appropriate lines/

Answer>

The p values lines in Tables 1 and 2 were modified.

  1. Lines 130-132: The authors should describe which patients were considered  with elderly onset and which, with early onset here. Description of clinical characteristics of patients with RA are badly required. This should be corrected.

Answer>

We mentioned the following sentences about the definition and clinical characteristics of elderly-onset RA and young-onset RA in 2.3 study end points and 1. introduction. Mentioned senteces are marked in red in the manuscript.  

(In the manuscript/2.3 study end points)

We evaluated all-cause mortality and cardiovascular mortality of patients with and without RA who underwent PCI. We also assessed outcomes in patient subgroups with elderly-onset RA (presentation after the age of 65 years) and young-onset RA (presentation before the age of 65 years).

(In the manuscript/ 1. introduction)

Recent studies demonstrated different features in elderly patients with RA, consisting of patients with elderly-onset RA and young-onset RA. Elderly-onset RA has a characteristic pattern with more comorbidities, a higher portion of male patients, acute onset, systemic involvement, and worse functional outcomes than young-onset RA [5,6].

  1. Lines 240-245: The authors should not repeat their Results at Discussion section. This should be corrected.

Answer>

Politely speaking you to the opinions of the authors, these sentences were left as sentences containing the author's analysis of the result rather than repeating the results. Instead, we deleted references to repeated results.

(In the manuscript/ 4. Discussion)

Late-onset RA was also an independent predictor of increased mortality (Table 4) compared with early-onset RA. Therefore, the higher component of late-onset RA (71.7%) in the RA subgroup may result in lower survival rates among patients with RA than those in the non-RA group.

  • Late-onset RA was also an independent predictor of increased mortality compared with early-onset RA. Therefore, the higher component of late-onset RA in the RA subgroup may result in lower survival rates among patients with RA than those in the non-RA group. (revised)

  1. Lines 269-272: This sentence contains contradictory statements. This should be corrected.

Answer> Considering the contradictory statements of the sentence you pointed out, we modified it as follows.

Although obesity is a traditional cardiovascular risk factor, it was a factor that reduced mortality in our study because a low or normal weight, indicative of a state of rheumatoid cachexia, may be associated with an increased risk of mortality due to muscle loss caused by inflammatory cytokines in RA.

  • Conversely, obesity was a factor that reduced mortality in our study because a low or normal weight, indicative of a state of rheumatoid cachexia, may be associated with an increased risk of mortality due to muscle loss caused by inflammatory cytokines in RA. (revised)

  1. Lines 244-247: This data is related to Introduction section. This should be corrected.

Answer>

Politely speaking you to the opinions of the authors, these sentences were left as linking sentences to mention the relationship between RA and cardiovascular risk in the previous text and present the relationship between RA and PCI outcomes in the following text. Please consider the opinions of the authors.

Reviewer 2 Report

Dear Authors,

I would like to thank you for having chance to review the manuscript: "Outcomes of Percutaneous Coronary Intervention in Elderly Patients With Rheumatoid Arthritis: A Nationwide Population- 3 Based Cohort Study".

My concerns are related to:

1.      significant differences between both presented groups. I would suggest propensity matching to obtain objective comparison.

2.     Please present all “P values for all parameters in Table 2 as the results are unclear for potential reader.

3.     The multivariable models for all-cause and CVD related mortality are multifactorial that raise the question of the main predictive parameter

4.     As you focused on age, please present ROC analysis to justify your decision.

Kind regards

Reviewer

Author Response

Thank you for your comments.

We did our best to respond to your comments.

We also discussed your comments with the statistician who participated in our study.

All changes are marked in red in the revised manuscript.

Reviewer 2’s comments

  1. significant differences between both presented groups. I would suggest propensity matching to obtain objective comparison.

Answer>

At the analysis stage, we considered the propensity matching method in consultation with statisticians. However, it was thought that there were no confounding variables that could affect the analysis results except for the gender between the two groups.The difference in gender distribution between groups was judged to be due to clinical characteristics according to the presence or absence of rheumatoid arthritis, and consequently, matching was not performed. Instead, we presented the results of multivariable logistic regression analyses for the association between clinical factors including gender and survival outcomes.

Please consider this.

  1. Please present all “P values for all parameters in Table 2 as the results are unclear for potential reader.

Answer>

Considering your opinion, P values are presented in all parameters of table 2. The modified Table 2 is attached.

  1. The multivariable models for all-cause and CVD related mortality are multifactorial that raise the question of the main predictive parameter

Answer>

In our study, multivariate logistic regression analysis was performed by controlling confounding variables, which were selected as traditional risk factors for cardiovascular disease, risk factors that can affect cardiovascular disease-related mortality, and adjustment variables related to PCI outcomes through previous studies

(reference:

  • Nochioka K, et al. Long-term outcomes in patients with rheumatologic disorders undergoing percutaneous coronary intervention: a BAsel Stent Kosten-Effektivitats Trial-PROspective Validation Examination (BASKET-PROVE) sub-study. European Heart Journal: Acute Cardiovascular Care 2017;6:778-786
  • SJ Ha, et al. Clinical outcomes of patients with rheumatoid arthritis whoc underwent percutaneous coronary intervention: A Korean nationwide cohort study. PLos ONE 18(2):e0281067
  • Wong B, et al. Very elderly patients with acute coronary syndromes treated with percutaneous coronary intervention.

To support the statistical analysis method of this study, journals that performed multivariate logistic regression analysis with selected risk factors are presented.

(reference:

  • Kronzer VL, et al. Lifestyle and clinical risk factors for incident rheumatoid arthritis-associated interstitial lung disease. The Journal of Rheumatology 2021;45:656-63
  • Kokkonen H, et al. Cardiovascular risk factors predate the onset of symptoms of rheumatoid arthritis: a nested case-control study. Arthritis research & Therapy 2017;19:148

  1. As you focused on age, please present ROC analysis to justify your decision.

Answer>

We agree with you and present ROC analysis.

Attached figure: Receiver-operating characteristic curve for all-cause (A) and cardiovascular (B) mortalities after PCI in elderly patients with RA. These curves were fitted using age and clinical factors (shown as combined in the figure) applied in multivariable logistic regression analysis. AUC : area under the curve; PCI: percutaneous coronary intervention; RA: rheumatoid arthritis.

Reviewer 3 Report

Title: Outcomes of Percutaneous Coronary Intervention in Elderly Patients with Rheumatoid Arthritis: A Nationwide Population Based Cohort Study.

Reviewer Comments: 

Authors tried to shed light on clinical outcomes of elderly patients with and without RA who underwent PCI. In this cohort study authors used database to extract data on patients aged ≥65 years who were diagnosed with coronary disease and underwent PCI. The primary outcome authors were looking was survival rate among elderly patients with and without RA. Authors observed that all-cause mortality survival rate was low in patients with RA than that in patients without RA. Authors also identified that elderly patients with RA who underwent PCI had a high mortality rate, particularly those with elderly onset RA than early onset RA.

Strength:

1.    One of the strengths of this study is that high sample size. (More than 14,000

Weaknesses:

1.    Figure 1 is not readable. It needs to be reformatted.

2.    It’s observational study and having physiological or clinical data would support the work.

3.    Figures 2 and 3 are not legible. It needs to be redrawn. Increase the font size on X and Y axis. 

4.    Are all these patients being at same stages of RA? Some may be at early stages of RA, even If they are elderly. Mortality rate would be the same If they are elderly and at early stages of RA? 

5. Is it possible to tailor RA treatment for patients with coronary disease? 

Author Response

Thank you for your comments.

We did our best to respond to your comments.

All changes are marked in red in the revised manuscript.

Reviewer 3’s comments

Title: Outcomes of Percutaneous Coronary Intervention in Elderly Patients with Rheumatoid Arthritis: A Nationwide Population Based Cohort Study.

Reviewer Comments: 

Authors tried to shed light on clinical outcomes of elderly patients with and without RA who underwent PCI. In this cohort study authors used database to extract data on patients aged ≥65 years who were diagnosed with coronary disease and underwent PCI. The primary outcome authors were looking was survival rate among elderly patients with and without RA. Authors observed that all-cause mortality survival rate was low in patients with RA than that in patients without RA. Authors also identified that elderly patients with RA who underwent PCI had a high mortality rate, particularly those with elderly onset RA than early onset RA.

Strength:

  1. One of the strengths of this study is that high sample size. (More than 14,000

Weaknesses:

  1. Figure 1 is not readable. It needs to be reformatted.

Answer>

Figure 1 has been reformatted for better viewing. We are attaching the revised figure 1 file.

  1. It’s observational study and having physiological or clinical data would support the work.

Answer>

  • Clinical data

Due to the low prevalence of RA and difficulties in enrolling elderly patients in clinical trials, clinical studies involving elderly rheumatoid arthritis patients are lacking. However, there is a clinical study that analyzed the prognosis of patients who underwent PCI for autoimmune disease (including systemic sclerosis, lupus, rheumatoid arthritis, and psoriatic arthritis ), although it was not mentioned in the manuscript because it was not a study targeting only patients with rheumatoid arthritis. This study was a post-hoc analysis study using a large, prospective, randomised, multicenter trials. In this study, patients with coronary artery disease who underwent PCI in the presence of rheumatic disease demonstrated a poor prognosis. 197 patients with autoimmune disease were included, and the average age was 68.7 ± 10.0 years, which is considered to be evidence to support the prognosis of elderly rheumatoid arthritis patients.

(Reference: Nochioka K, et al. Long-term outcomes in patients with rheumatologic disorders undergoing percutaneous coronary intervention: a BAsel Stent Kosten-Effektivitats Trial-PROspective Validation Examination (BASKET-PROVE) sub-study. European Heart Journal: Acute Cardiovascular Care 2017;6:778-786)

  • Physiological data

We present additional journal describing the mechanisms by which cardiovascular disease increases in patients with rheumatoid arthritis.

(Reference: England BR, et al. Increased cardiovascular risk in rheumatoid

arthritis: mechanisms and implications. BMJ 2018;361 k1036)

  1. Figures 2 and 3 are not legible. It needs to be redrawn. Increase the font size on X and Y axis. 

      Answer>

Figures 2 and 3 have been modified again to make it more visible. The font size of the X and Y axes has also been increased. We are attaching the revised figure 2 and 3 files.

  4. Are all these patients being at same stages of RA? Some may be at early       stages of RA, even If they are elderly. Mortality rate would be the same If they are elderly and at early stages of RA? 

Answer> I agree with your question. From this point of view, our authors assessed survival outcomes in RA subgroups divided into patients with elderly-onset RA and young-onset RA. As you mentioned, some may have early stages of RA even at an advanced age. These patients may be classified as late-onset RA (presentation after the age of 65 years). In our study, the all-cause and cardiovascular disease-associated survival rates were lower in the late-onset RA group compared with those in non-RA and early-onset RA groups. As mentioned in the manuscript, patients with late-onset RA often have comorbidities and high disease activity of rheumatoid arthritis, which may have contributed to the poor survival outcomes.

  1. Is it possible to tailor RA treatment for patients with coronary disease? 

Answer>

The authors answer that it is possible.

  • The treatment recommendations of the European League Against Rheumatism, updated and published in 2015/2016, emphasize the risk of cardiovascular disease (CVD) in patients with rheumatoid arthritis and suggest treatment strategies to reduce the risk of cardiovascular disease This recommendation was made based on the formation of a multidisciplinary steering committee that included not only rheumatologist but also cardiologists and epidemiologists. This recommendation included CVD risk estimation, CVD risk prediction models, CVD risk management, and medication use of rheumatoid arthritis to reduce the risk of cardiovascular disease in patients with rheumatoid arthritis. If this recommendation is well applied to elderly RA patients who underwent PCI, which was suggested to have a poor prognosis in our study, we believe that tailored RA treatment is possible for patients with coronary artery disease.

(Reference: Agca R, et al. EULAR recommendations for cardiovascular disease risk management in patients with rheumatoid arthritis and other forms of inflammatory joint disorders: 2015/2016 update. Ann Rheum Dis 2017;76:17-28)

  • Other journal also suggest treatment strategies such as multifaceted, team approach multi-team to target CVD risk reduction in rheumatoid arthritis

(Reference: England BR, et al. Increased cardiovascular risk in rheumatoid

arthritis: mechanisms and implications. BMJ 2018;361 k1036)

Round 2

Reviewer 2 Report

Dear Authors,

I would like to thank you for having chance to review the improved version of your manuscript after corrections.

I would suggest adding a separate section for “conclusion” as in present form it’s just a continuation of discussion. Please, as a separate section (5).

The conclusion is not clear for potential reader as the “elderly” should be clarified presenting the cut of value of the patients’ age. The same suggestion refers to “late-onset RA”, please present the age cut of value for late RA diagnosis that has a potential prognostic value.

Kinds

Tom

Author Response

Thank you for your comments.

All changes are marked in red in the revised manuscript. 

Reviewer 2’s comments

I would suggest adding a separate section for “conclusion” as in present form it’s just a continuation of discussion. Please, as a separate section (5).

The conclusion is not clear for potential reader as the “elderly” should be clarified presenting the cut of value of the patients’ age. The same suggestion refers to “late-onset RA”, please present the age cut of value for late RA diagnosis that has a potential prognostic value.

Answer>

We added section of 5. Conclusions.

Considering your comment, we have corrected the sentence to clarify the age cut of value.

(Manuscript)

In conclusion, elderly patients with RA who underwent PCI had an increased risk of all-cause and cardiovascular mortalities; this trend was more specifically observed in patients with elderly-onset RA than in those with young-onset RA. Patient characteristics and clinical factors associated with the survival outcomes need to be considered during the treatment of elderly paitents with RA for a better prognosis of cardiovascular diseases.

->

  1. Conclusions

Elderly patients aged ≥ 65 years with RA who underwent PCI had an increased risk of all-cause and cardiovascular mortalities; this trend was more specifically observed in patients with elderly-onset RA who presented after the age of 65 years than in those with young-onset RA who presented before the age of 65 years. Patient characteristics and clinical factors associated with the survival outcomes need to be considered during the treatment of elderly paitents with RA for a better prognosis of cardiovascular diseases
